# The Human Amniotic Membrane: A Rediscovered Tool to Improve Wound Healing in Oral Surgery

**DOI:** 10.3390/ijms26178470

**Published:** 2025-08-31

**Authors:** Maurizio Sabbatini, Paolo Boffano, Martina Ferrillo, Mario Migliario, Filippo Renò

**Affiliations:** 1Sciences and Innovative Technology Department, Università del Piemonte Orientale, Viale T. Michel 11, 15121 Alessandria, Italy; maurizio.sabbatini@uniupo.it; 2Traslational Medicine Department, Università del Piemonte Orientale, Via Solaroli n. 17, 28100 Novara, Italy; paolo.boffano@uniupo.it (P.B.); mario.migliario@med.uniupo.it (M.M.); 3Department of Health Sciences, University of Catanzaro “Magna Graecia”, 88100 Catanzaro, Italy; martina.ferrillo@unicz.it; 4Health Sciences Department, Università of Milan, Via A. di Rudini n. 8, 20142 Milano, Italy

**Keywords:** oral mucosa, wound healing, oral surgery, staminal cells, biomaterials, growth factors

## Abstract

Wound healing in oral surgery is influenced by systemic conditions (aging, diabetes) and habits (smoking, alcoholism), which can hinder the natural regenerative capacity of the oral mucosa. The human amniotic membrane (hAM), long recognized for its wound-healing properties, has gained attention as a valuable biomaterial in regenerative dentistry. Its biological composition—including epithelial and mesenchymal stem cells, collagen, growth factors, cytokines, and proteins with anti-inflammatory and antimicrobial properties—supports anti-inflammatory, angiogenic, immunomodulatory, and pro-epithelializing effects. These elements work synergistically to enhance tissue repair, reduce scarring, and promote rapid healing. The hAM can be preserved through cryopreservation, dehydration, or freeze-drying, maintaining its structural and functional integrity for diverse clinical uses. In oral surgery, the hAM has been applied with significant success to surgical wound coverage, treatment of periodontal and bone defects, and implant site regeneration, as well as management of complex conditions like medication-related osteonecrosis of the jaw (MRONJ). Clinical studies and meta-analyses support its safety, efficacy, and adaptability. Despite its proven therapeutic benefits, the hAM remains underutilized in dentistry due to challenges related to its preparation and storage. This review aims to highlight its potential and encourage broader clinical adoption in regenerative oral surgical practices.

## 1. Introduction

Wound healing in oral surgery is a multifactorial and complex process. It ensures the successful recovery of tissues following surgical procedures such as extractions, periodontal surgeries, implant placements, and other oral interventions, being crucial to maintaining the integrity of oral anatomical structure and function [1]. The oral cavity exhibits a remarkable capacity for healing, primarily attributed to its rich blood supply and to the presence of saliva [2]. Despite its exposure to mechanical tension, abrasion, and different microbial populations, the oral mucosa demonstrates effective regenerative abilities [3]. In fact, the recent literature suggests that certain microbic populations may positively influence wound healing by modulating the immune system response [3,4,5,6]. Additionally, the oral mucosa possesses a notably thicker epithelial layer compared to skin—comprising approximately 20–30 living cell layers in the basal lamina versus 5–8 in skin—along with a higher cellular proliferation rate, which further facilitates rapid tissue repair [7]. Moreover, the buccal mucosa also exhibits mechanical flexibility, withstanding both stretch and compression during functional activities [8,9]. However, intrinsic factors such as aging, diabetes, and psychological stress, along with extrinsic factors like alcohol consumption and cigarette smoking, have been shown to impair the healing process in the oral mucosa [10,11]. Non-healing wounds may present long-term or excessive inflammation, persistent infection, and microbial biofilm formation, eventually leading to permanent damage to the physiological function and appearance of the oral region.

Problematic wound-healing processes may be detrimental for the success of surgical intervention as well as for the quality of life of the patient. Therefore, improvement in wound healing after oral surgery interventions in specific conditions could be critical to achieve favorable clinical outcomes [12]. Several innovative methods and medications have been tested in the last fifteen years to improve wound healing, both during and after oral surgery, typically, cell-based therapy or biomaterial-alone therapy [13]. Lately, the use of the amniotic membrane has emerged as a versatile and promising resource for regenerative medicine, ophthalmology, and wound care [1]. Its multifaceted properties, ranging from anti-inflammatory and antimicrobial attributes to wound-healing capabilities, have catapulted it into the spotlight of medical research and clinical applications [14].

In oral surgery, the utilization of graft materials has been extensively studied and implemented [14]. However, both absorbable and nonabsorbable membranes currently used for regenerating soft and hard oral tissues present certain limitations [15,16]. Nonabsorbable membranes are frequently associated with complications such as oral exposure through soft tissue, which necessitates a second surgical procedure for membrane removal. On the other hand, resorbable membranes tend to have low mechanical strength, and their degradation can trigger a significant inflammatory response during healing. Additionally, these membranes lack inherent biological properties, prompting the search for more effective alternatives [13,17].

In this context, placental membranes have been suggested as promising bioactive materials for guided tissue regeneration in the oral cavity, offering potential advantages over traditional membrane types [18,19,20,21]. In particular, the hAM has been proposed to improve wound healing in periodontal surgery, oral reconstructive surgery after tumor resection or oral lesion excision, oronasal fistula repair, prosthodontic surgery, and finally in medication-related osteonecrosis of the jaw (MRONJ) [1,14,22,23].

## 2. Human Amniotic Membrane’s Structural Complexity

The human amniotic membrane (hAM) is the innermost layer of the placenta, distinguished by its thin, semi-transparent, and somewhat rough texture, with thickness ranging from 0.02 mm to 0.5 mm [24,25,26].

Notably, the hAM lacks blood vessels and nerves, relying on diffusion from chorionic fluid, amniotic fluid, and fetal surface vessels to obtain nutrients and oxygen [24,27,28]. Due to the limited oxygen supply, energy production predominantly occurs through anaerobic glycolysis [24,27]. Structurally, the hAM comprises five distinct layers: an epithelial monolayer, a single layer of epithelial cells; a basement membrane that provides structural support to the epithelial layer; a compact layer, a dense connective tissue layer; a fibroblast layer, containing fibroblast cells responsible for producing extracellular matrix components; and an intermediate/spongy layer consisting of a looser, sponge-like connective tissue layer [29]. This multilayered architecture contributes to the membrane’s unique physical and biological properties, making it valuable in various clinical and regenerative applications.

The epithelial monolayer, situated closest to the fetus, plays a crucial role in active secretion and transport functions. This layer consists of a single sheet of cuboidal epithelial cells firmly anchored to the basement membrane [27,28]. Beneath this epithelium lies a condensed acellular layer composed of collagen types I, II, and V [24]. The amniotic epithelial cells are characterized by numerous microvilli on their apical surface, which enhance their secretory activity. These cells produce a variety of growth factors, including epidermal growth factor (EGF), basic fibroblast growth factor (bFGF), vascular endothelial growth factor (VEGF), transforming growth factor alpha and beta (TGF-α/β1-3), and platelet-derived growth factor (PDGF) [30]. Additionally, they facilitate intra- and transcellular transport processes, supporting membrane exchange and material transfer essential for fetal development [27,28].

The basement membrane is one of the thickest membranes of all human tissues, and it is composed of collagen types IV and V, fibronectin, laminin, and nidogen. The basement membrane is characterized by a rich composition of proteoglycans, predominantly containing heparan sulfate, which function as a permeable barrier to amniotic macromolecules. Additionally, it contains several structural molecules crucial for maintaining membrane integrity. These include actin, α-actinin, spectrin, ezrin, various cytokeratins, vimentin, desmoplakin, and laminin [26,31]. Laminin, in particular, has been extensively studied due to its multifaceted role in cellular functions. It contributes to cell survival, differentiation, shape, and motility, and plays a vital role in maintaining tissue phenotypes [26,32]. The compact layer lies adjacent to the basement membrane, and it represents the fibrous skeleton of the amnion. The fibroblastic layer is responsible for secretion of collagen types I, III, and VI, which help to maintain the mechanical integrity of the membrane [26].

The amnion (AM) is a vital component of the fetal membranes, featuring a complex structure and multiple functions. Its outer layer consists of mesenchymal fibroblast-like cells, likely originating from the mesodermal embryonic plate, dispersed throughout a full-term membrane. This mesenchymal layer is rich in collagen, which enhances the tensile strength of the membrane [26,33]. The outermost layer is referred to as the “zona spongiosa” due to its high content of proteoglycans and glycoproteins, imparting a spongy appearance under the microscope [26,33] (Figure 1).

Beyond being a simple, avascular barrier, the amnion performs several metabolic roles, including transporting water and soluble substances, as well as producing bioactive molecules such as vasoactive peptides, growth factors, and cytokines. Its primary functions involve protecting the developing embryo from dehydration and creating a supportive environment that allows for free growth, free from external pressures [26,33].

The mechanical properties of the amnion are largely determined by its collagen composition. Its resistance to tractional forces stems mainly from a condensed layer of interstitial collagen types I and II, along with elastin. Conversely, the membrane’s elasticity is primarily due to collagen type III. The presence of interstitial collagens also confers resistance to proteolytic enzymes, protecting the amnion from enzymatic degradation [24,28,33].

## 3. The hAM in Tissue Repair

The historical use of the amniotic membrane dates back centuries, with ancient civilizations recognizing its efficacy in wound healing and as a biological dressing [22]. However, it was in the 20th century that scientific advancements, particularly in tissue engineering and regenerative medicine, reignited interest in this biological component [22,33]. The discovery of its immunomodulatory and anti-inflammatory properties ushered in a new era of exploration, leading to its incorporation into diverse medical procedures [34,35].

One of the most notable applications of the amniotic membrane resides in ophthalmology, where its transparency, biocompatibility, and capacity to promote epithelialization have proven invaluable [36]. Ocular surface disorders, corneal ulcers, and various ocular pathologies have become prime targets for intervention using amniotic membrane grafts. The membrane’s ability to foster corneal healing, diminish inflammation, and reduce scarring has positioned it as a vital tool in the armamentarium of ophthalmic surgeons [36]. This success has brought physicians to further explore the amniotic membrane’s regenerative potential in diverse medical fields, including orthopedics, dermatology, and plastic surgery [37]. Its capacity to expedite wound healing, modulate inflammatory responses, and foster tissue regeneration positions it as an invaluable asset in the treatment of chronic wounds, burns, and soft tissue injuries. The extracellular matrix within the membrane provides a scaffold for cellular migration and proliferation, orchestrating the repair of damaged tissues [38].

The clinical applications of the amniotic membrane continue to burgeon, with ongoing research exploring its potential in cutting-edge areas such as tissue engineering, stem cell therapy, and immune modulation [39]. As understanding deepens regarding its molecular and cellular properties, new therapeutic avenues are anticipated to emerge. Challenges, including the standardization of processing methods and regulatory considerations, are being actively addressed to ensure the safety and efficacy of amniotic membrane-based therapies [40].

The hAM has garnered significant interest in the field of oral surgery due to its unique biological properties and potential to address several unmet clinical needs. As a biological tissue derived from the amniotic sac of the placenta, the hAM offers a promising alternative or adjunct to traditional techniques for tissue regeneration, wound healing, and management of complex oral defects [33,38,40].

### Stem Cell Characteristics of Amnion-Derived Cells

The human amniotic membrane contains two primary populations of stem cells: amniotic epithelial cells (AECs), which rest on a basement membrane, and amniotic membrane stem cells (AMSCs), located within the fibroblast layer of the membrane [41,42] (Figure 1). Both cell types possess embryological-origin characteristics and are generated prior to the formation of the three germ layers during pre-gastrulation stages [43]. Specifically, AECs originate from the pluripotent epiblast on the eighth day of conception, whereas AMSCs derive from the extraembryonic mesoderm of the primitive streak [44].

hAM-derived cells are characterized by a lack of immunogenicity, meaning that they are less likely to provoke an immune response. This trait contributes to their immunological tolerance, making them suitable for various therapeutic applications without significant risk of rejection [45]. Furthermore, AECs and their conditioned medium have demonstrated a capacity to substantially inhibit T-cell proliferation. Notably, this suppression occurs regardless of direct cell contact, indicating that soluble factors secreted by AECs play a significant role in modulating the immune response [46]. Recent studies suggest that AECs can create a supportive microenvironment that promotes cell survival and endogenous tissue regeneration. They achieve this by secreting a range of bioactive molecules, including cytokines, growth factors, and exosomes, which collectively facilitate tissue repair and regeneration [47].

Several studies have highlighted that AECs and AMSCs can differentiate into all three germ layers—the endoderm, mesoderm, and ectoderm—and consequently develop into specific cell types such as chondroblasts, osteoblasts, adipocytes, myocytes, and neuronal cells [29]. In fact, both epithelial and mesenchymal amniotic cells express several surface markers associated with embryonic stem cells, including SSEA-3, SSEA-4, TRA-1-60, and TRA-1-81 [37]. Cells also express pluripotency markers such as OCT-4, HNF-3β, nanog, and nestin, indicating their inherent potential for differentiation into multiple cell lineages [48,49,50,51].

In particular, the potential anticancer effects of AMSCs have also been highlighted in both in vivo and in vitro studies. They secrete exosomes and extracellular vesicles that possess immunomodulatory properties, promoting tissue regeneration and offering therapeutic benefits in preclinical disease models. Collectively, these findings point to the promising use of AMSC-derived extracellular vesicles as cell-free therapeutic agents for cancer and immune-related conditions [33,52].

Stem cell therapy is increasingly recognized as a promising approach for tissue regeneration, especially in the context of periodontal repair. These therapies leverage the high proliferative capacity, ability to differentiate into various cell types, and functional properties of stem cells to promote healing of damaged tissues. When incorporated into periodontal wounds, stem cells, particularly AMSCs, can enhance regenerative processes by modulating inflammation, promoting new tissue formation, and accelerating overall healing [1].

Current research efforts are focused on developing bioengineered wound-healing products that incorporate AMSCs. These innovative approaches aim to improve clinical outcomes in periodontal therapy, offering potential for more effective and faster tissue regeneration [33,51]. The integration of stem cell-based strategies into periodontal treatment protocols holds great promise for advancing regenerative dentistry and improving patient care.

## 4. hAM’s Multifaceted Properties

### 4.1. Biomechanical Properties

The hAM is characterized by great elasticity, due to the presence of elastin, proteoglycan, and collagen [53], which, together with its thickness included between 0.02 and 0.5 millimeters, provide noticeable strength and robustness for several clinical uses. Furthermore, the hAM has shown great resistance to various proteolytic factors owing to the presence of interstitial collagens. The self-adhesive property of the hAM was highlighted too: it may adhere to the recipient’s exposed root on gingival recessions, thus eliminating the need for suturing and significantly reducing the operation time (as it does not require a second surgical site) [33,40] (Figure 2).

### 4.2. Epithelialization

Oral wound healing involves a complex interplay of physical and molecular processes that facilitate tissue repair, starting with hemostasis and inflammation, followed by proliferation, where new tissue and blood vessels form, and culminating in remodeling to restore tissue integrity; these stages are driven by cellular activities such as keratinocyte migration, fibroblast proliferation, and immune cell response, all coordinated through signaling molecules like cytokines, growth factors, and extracellular matrix components, ensuring effective recovery of oral mucosal integrity. The hAM may act as a basement membrane that facilitates epithelialization by aiding epithelial cell migration, epithelial differentiation, basal cell adhesion, and epithelial apoptosis prevention [54,55] (Figure 2).

The hAM produces various growth factors that can promote epithelialization. These include the mRNA expression of epidermal growth factor (EGF), keratinocyte growth factor (KGF), hepatocyte growth factor (HGF), basic fibroblast growth factor (bFGF), and transforming growth factors (TGF-α, TGF-β1, TGF-β2, TGF-β3). Additionally, growth factor receptors such as KGFR and HGFR are present in both stromal and epithelial regions of the amnion [40]. Furthermore, the hAM promotes healing and wound epithelialization while reducing granulation tissue formation [40]. A prospective clinical trial involving patients with chronic venous leg ulcers (5–25 cm^2^) demonstrated successful wound healing from the edges following treatment with cryopreserved allogenic human amnion membranes. These ulcers had previously shown no tendency to heal and were resistant to standard medical therapies. Over a 3-month follow-up period, 80% of patients experienced at least a 50% reduction in ulcer size. No adverse effects were reported, supporting the safety and reliability of cryopreserved amnion membranes for clinical use [56].

### 4.3. Inhibition of Inflammation

The recruitment of specific inflammatory immune cell subsets, such as neutrophils, macrophages, and T cells, plays a crucial role in orchestrating the wound-healing process by clearing pathogens, removing cellular debris, and secreting cytokines and growth factors that promote tissue regeneration and re-epithelialization; however, an imbalance or prolonged presence of these cells can lead to chronic inflammation and impaired healing, highlighting the importance of tightly regulated immune responses for optimal tissue repair. The hAM is believed to decrease inflammatory cell influx to the wound area and consequently to decrease inflammatory mediators by serving as a barrier. Nevertheless, the anti-inflammatory properties of the amniotic membrane are not clear yet [33,40].

For example, interleukin-1*α* and interleukin-1*β*, both proinflammatory mediators, are suppressed by the matrix of stroma of the hAM. Likewise, the hAM presents great quantities of hyaluronic acid, which acts as a ligand for CD44 which is expressed on inflammatory cells. It also expresses secretory leukocyte proteinase inhibitor and elafin; these inhibitors have both anti-inflammatory and antimicrobial properties. It is also thought that the hAM causes the downregulation of pro-inflammatory cytokines, such as TNF-α and IL-6, and activation of neutrophils and M1 and M2 macrophages, which help to relieve pain. Its stromal matrix also shows a marked suppression of proinflammatory cytokine, IL-1α, and IL-1β expression [57].

### 4.4. Angiogenesis

Angiogenesis and neovascularization are critical processes during the proliferative phase of wound healing, as they facilitate the formation of new blood vessels to supply oxygen and nutrients essential for tissue repair. These processes contribute significantly to granulation tissue development, comprising approximately 60% of its mass in the early stages, thereby supporting the regeneration of damaged tissue and promoting effective wound closure [58].

The proangiogenic potential of AMSCs (Adipose-derived Mesenchymal Stem Cells) has been extensively studied, highlighting their ability to promote new blood vessel formation. This capacity is largely attributed to their high expression levels of key proangiogenic genes such as vascular endothelial growth factor-A (VEGF-A), angiopoietin-1, HGF, and fibroblast growth factor-2 (FGF-2), as well as antiapoptotic proteins like protein kinase-Bα (known also as AKT-1). Experimental administration of AMSCs in models of hindlimb ischemia in mice has demonstrated significant improvements, including increased blood flow and higher capillary density. These findings underscore the potential of AMSCs to enhance neovascularization and tissue repair in ischemic conditions [59] (Figure 2).

### 4.5. Inhibition of Scarring

Collagen synthesis is essential for effective tissue regeneration, as it underpins the structural integrity of healed tissues, facilitates matrix remodeling, and provides a resilient framework that shields the wound from mechanical stresses, thereby ensuring proper matrix maturation and overall tissue functionality [60,61,62].

The hAM decreases the risk of fibrosis by the downregulation of TGF*β*, modulated by hyaluronic acid, and its receptor expression by fibroblasts. Thanks to this property and to the differentiation inhibition of fibroblasts into myofibroblasts, scarring is reduced by the hAM while it modulates wound healing by promoting the reconstruction of tissues. Moreover, the hAM may decrease protease activity due to the secretion of tissue inhibitors of metalloproteinases (TIMPs) [40,63,64].

### 4.6. Lack of Immunogenicity

Human leukocyte antigens (HLAs) are crucial components of the immune system, involved in presenting peptides to immune cells. In the hAM, HLA Class I molecules are expressed on amniotic epithelial and mesenchymal cells, which means that these cells have the capacity to present endogenous peptides to immune cells, playing a role in immune tolerance and immune modulation during pregnancy. However, HLA Class II molecules are not synthesized by these cells, indicating that amniotic membrane cells do not present exogenous antigens in the same way professional antigen-presenting cells do. The lack of classical HLA Class II expression reduces direct aggressive T-cell recognition, lowering immune rejection risk, inducing a Th2 cytokine bias, and promoting regulatory immune cells (like Myeloid-derived suppressor cells (MDSCs) or regulatory T cells), supporting immune tolerance to administered antigens [65]. This immunological environment is associated with clinical safety as it minimizes immune rejection and inflammation against orally delivered agents [66].

This differential expression contributes to the immune-privileged status of the amniotic membrane, making it a valuable tissue in transplantation and regenerative medicine due to its low immunogenicity [67].

In summary, the clinical safety of oral administration relating to rejection and immune tolerance is underpinned by mechanisms observed in pregnancy, where the specialized modulation of HLA molecules and the immune environment promote tolerance rather than rejection. This includes the suppression of classical HLA Class II expression, promotion of Th2 immunity, and activation of suppressive immune cells, collectively creating a tolerogenic milieu that minimizes immune-mediated damage [68]. This natural tolerance model informs strategies to enhance safety in oral immunomodulatory therapies [69] (Figure 2).

### 4.7. Antimicrobial/Antiviral Properties

The hAM may prevent infiltration and adhesion of microorganisms to wound surfaces by acting as a barrier. Furthermore, it produces *β*-defensin, which is part of the adaptive immune response mechanisms [70], with the predominant type, present in the amniotic epithelium, being *β*3-defensin (Figure 2).

Epithelial and mesenchymal amniotic cells release proteins like Activin A, IL-1 receptor antagonist (IL-1ra), and IL-10, which are incorporated into the amniotic membrane stroma, contributing to anti-inflammatory effects [34,71]. Moreover, the amniotic membrane inhibits proteinases and matrix metalloproteinases (MMPs), reducing infiltration of inflammatory cells into tissues [72]. The amnion cells can promote apoptosis in leukocytes, aiding in resolving inflammation. This is facilitated by apoptosis-inducing genes such as Fas L, tumor necrosis factor (TNF), and TNF-Related Apoptosis-Inducing Ligand (TRAIL) expressed by amniotic epithelial cells [1]. Kanyshkova and coll. reported the presence of the antibacterial protein lactoferrin in the membrane [73]. Recent studies indicate that a cryopreserved hAM can effectively prevent wound-related infections due to its rich content of growth factors and anti-inflammatory components [74].

Instead, the presence of cystatin E, the analog of cysteine proteinase inhibitor, gives the hAM antiviral properties. Currently, several studies are investigating the antiviral effects of the hAM; it can exert antiviral properties by either limiting virus colonization and replication or mitigating the severe consequences resulting from an abnormal host immune response to infection [75]. For example, in individuals with herpes simplex virus type 1 (HSV-1) keratitis, hAM transplantation has been shown to promote rapid epithelial healing while reducing stromal inflammation and ulceration. These beneficial effects of the hAM may be attributed, at least in part, to its secretion of IL-1 receptor antagonist (IL-1RA), a natural inhibitor of IL-1α, which plays a key role in inflammatory processes [76].

Finally, two key features of the hAM help reduce bacterial load and infection risk by preventing microbial accumulation: first, the hemostatic property of collagen fibers, which helps prevent hematoma formation in clean surgical wounds; and second, its ability to adhere to the wound surface, which prevents dead space formation and accumulation of serous discharge [1,62].

### 4.8. Aesthetic Properties

The hAM provides excellent aesthetic results for texture and color match to the recipient site. Due to its aesthetic characteristics, the hAM could be one of the options considered in oral cavity periodontal procedures [77,78]. Its cosmetic applications showed rapid improvement in midface aging correction cases, including filling the nasolabial folds and malar fat pad [62,78].

### 4.9. Reducing Pain at the Site of Application

The application of the hAM to wounds significantly decreases the pain experienced by patients. This pain reduction is thought to result from the membrane’s ability to adhere to the wound site, covering exposed nerve endings and thereby providing a protective barrier [56,79]. Additionally, hAM adherence prevents direct contact between the lesion and external contaminants, while its porous structure facilitates the evaporation of wound fluid. These combined effects help reduce plasma loss, which can promote a healthier healing environment, and are believed to contribute to the prevention of infection and sepsis in the treated wounds [80]. Furthermore, it has been observed that the soft mucoid lining of the hAM protects the exposed nerve endings from external irritants, helping to decrease pain sensations [62,81,82].

## 5. Preparation and Utilization of hAM

The hAM is typically harvested from the placenta, with the preferred method being during a planned cesarean section. This approach allows for the collection of the hAM under strict aseptic conditions, minimizing the risk of contamination. In contrast, vaginal delivery poses a higher risk of contamination due to the exposure to the birth canal’s microbial environment, making it less ideal for sterile harvesting of the hAM. The tissue is then subjected to additional decontamination through repeated washes with sterile saline or an established cocktail of antibiotics and antimycotics. This practice ensures the quality and safety of the membrane for its various medical applications [33,51].

Amniotic tissue is available in various commercial forms designed for different clinical applications. These include fresh-frozen injectable amniotic liquid, containing viable amniotic cells and/or particulate amniotic membranes, used for injection to promote healing; micronized freeze-dried (lyophilized) particulate powder, a dehydrated form that can be directly applied to wounds or reconstituted for injection, facilitating versatile use in wound management; and cross-linked dehydrated membranes, used as an adhesion barrier, providing a physical barrier to prevent tissue adhesions post-surgery or injury. These different formulations leverage the biological properties of amniotic tissue to promote healing, reduce inflammation, and prevent adhesions [51,83].

### 5.1. Cryopreservation

Cryopreservation is a widely used method for the treatment and storage of the hAM. This process involves placing the hAM in a storage solution—typically a mixture of 86% glycerol in Dulbecco’s Modified Eagle Medium (DMEM), supported by a carrier substrate—and storing it at −80 °C. Stable storage at this temperature generally allows the hAM to be preserved for up to 1 to 2 years, after which its efficacy may decline [83].

However, this method presents some challenges. One concern is the potential damage to the tissue’s integrity caused by crystallization within the tissue during freezing, which can compromise the hAM’s biological effectiveness. Specifically, crystallization may lead to reduced levels of essential growth factors and antiangiogenic intermediates critical for therapeutic outcomes.

Additionally, the use of a cryopreserved hAM necessitates a highly organized and reliable logistics system. Such a system must ensure continuous, stable storage at −80 °C across various stages—from initial preservation to transportation and eventual clinical use—to maintain the tissue’s quality and safety [84]. This complexity underscores the importance of meticulous planning and management in cryopreservation practices for the hAM.

### 5.2. Lyophilization

During the lyophilization process, the hAM is rapidly frozen to temperatures as low as −80 °C, and then subjected to high vacuum to remove water through sublimation, reducing water content to 5–10%. This process inhibits enzymatic activity and sterilizes the tissue via gamma irradiation, eliminating the need for cold storage and addressing logistical challenges of cryopreservation [1,85]. There is debate over whether standard lyophilization is beneficial or harmful compared to cryopreservation. While it offers logistical advantages, some studies report that gamma irradiation during sterilization causes significant damage to the tissue’s epithelium, basement membrane, and lamina densa, especially at higher doses (20–30 kGy). Radiation can destroy crucial cytokines involved in wound healing, epithelialization, reducing fibrosis and scarring, and promoting angiogenesis [86,87]. The use of trehalose as a lysoprotectant can help preserve tissue integrity. A trehalose-treated lyophilized hAM retains essential proteins (various collagen types, laminin-5, fibronectin) and maintains the native characteristics of the amniotic membrane, potentially improving therapeutic outcomes [88].

Finally, lyophilization or freeze-drying offers a cost-effective method for preserving the hAM, reducing storage expenses and facilitating easier use during surgeries [89]. These advantages, combined with the ability to implement safety protocols for preserved tissues, make lyophilization a practical option for maintaining the functional integrity of the amnion over extended periods, even at room temperature.

### 5.3. Dehydration

The dehydration protocol is an advanced method designed to preserve the biological properties of the hAM while minimizing tissue damage. Traditionally, the hAM can be dried under a biohazard hood by exposing it to air, followed by sterilization using gamma irradiation. While gamma sterilization effectively eliminates pathogens and negates the need for freezing, it may compromise tissue integrity due to radiation exposure [1,85].

Recently, a more refined technique involving low-temperature vacuum evaporation has been developed to dehydrate the hAM more gently. This method entails placing the amniotic membrane in a vacuum environment and drying it at a temperature between 3.5 °C and 6 °C after removing the spongy layer. Prior to drying, the tissue is pre-treated with raffinose—a sugar that helps stabilize cellular structures—and incubated with broad-spectrum antimicrobials to ensure sterility [57]. This innovative approach aims to better preserve the biochemical and structural properties of the tissue, maintaining its functional integrity for therapeutic applications.

The use of lysoprotective saccharides such as trehalose or raffinose, combined with the antioxidant epigallocatechin prior to drying, offers significant benefits in preserving the qualities of the hAM as a transplant material [90]. This approach helps maintain many of the characteristics of a fresh hAM, which contribute to its appeal in transplantation. Specifically, it has been shown to retain the biochemical stability of various growth factors and enzymes that are vital for the antifibrotic and anti-inflammatory effects of the hAM [90]. Additionally, this method produces a more stable hAM that can be stored at ambient temperatures, effectively overcoming the logistical challenges associated with cold storage during distribution.

The dehydrated hAM can also be micronized. This process allows it to be administered as a topical powder or mixed with saline to create an injectable solution or a topical gel. Use of the amniotic membrane has recently increased clinically as an allograft material for chronic and acute wound care management, for scar tissue reduction, as a barrier membrane, and as a soft tissue regeneration graft. The amniotic membrane is highly useful and effective as a culture substrate [91].

The selection of preservation methods is crucial and should align with the specific medical application and available resources. Ongoing research into various preservation protocols aims to enhance the quality and functionality of the preserved hAM. Advances in this field are promising, as they can lead to improved hAM-based products with greater efficacy and safety for clinical and regenerative medicine applications. Continuous innovation and optimization are essential to fully realize the potential of the hAM in various therapeutic contexts (Figure 3).

## 6. The hAM in Oral Surgery

Despite the oral mucosa’s remarkable ability to heal quickly and with minimal scarring, surgical and therapeutic interventions on this tissue can still lead to various problems.

Debelian and coll [92] conducted a study involving twenty-six patients, where blood samples taken ten minutes post-endodontic therapy revealed the presence of anaerobic bacteria and other oral microorganisms, indicating that bacteria from infected root canals can enter the bloodstream and potentially disseminate to vital organs such as the lungs, heart, and peripheral capillaries, highlighting the importance of infection control during dental procedures [92,93]. Following cleft palate surgery or tumor resection, oronasal fistulas (ONFs) can develop in up to 60% of cases due to factors like infection, flap necrosis, and tension, with cellular grafts offering structural and functional support to promote healing [94]. However, wound-healing complications are exacerbated by underlying autoimmune diseases such as pemphigoid, pemphigus vulgaris, and diabetes, as well as external factors like alcohol and smoking, leading to delayed healing and scarring [10,95,96].

Tissue graft therapies are promising but face limitations, including low cell graft viability, poor transmucosal permeability, and grafting difficulties, which hinder their effectiveness in achieving optimal oral wound repair [97,98]. These problems have encouraged clinicians to search for a valid system to promote oral mucosa healing, avoiding complications. The use of the hAM for the treatment of oral mucosa defects was initially described in 1985 by Lawson and coll [99]. The hAM is a promising grafting material for tissue reconstruction due to its multifaceted benefits. It supports tissue regeneration, reduces scarring, facilitates revascularization, and provides excellent wound coverage. This leads to improved wound healing, better postoperative function, and enhanced aesthetics, all with a low complication rate. hAM could be a viable option for a variety of reconstruction procedures [1].

The amniotic membrane aligns well with both the traditional mechanical concept of guided tissue regeneration (GTR) and the modern biological approach. In biomechanical GTR, the amniotic membrane not only preserves the structural and anatomical integrity of the regenerated tissues but also actively promotes healing by reducing postoperative scarring and preventing functional loss. Its rich source of stem cells further enhances its regenerative potential [1].

The amniotic membrane improves gingival wound healing, minimizes scarring, and demonstrates excellent revascularization, making it a highly effective grafting material for wound coverage. Its ability to promote wound healing, maintain postoperative function, and support aesthetic outcomes without complications makes the hAM a promising option for reconstructing oral cavity defects. Overall, the hAM offers a combination of biological and mechanical benefits that can facilitate successful tissue regeneration and functional restoration in oral surgeries [1,33].

Periodontal plastic surgical procedures focus on covering exposed root surfaces to restore periodontal health and aesthetics. Traditionally, obtaining sufficient graft material has involved harvesting tissue from a second surgical donor site, which can increase patient discomfort and surgical complexity. To address these challenges, various alternative additive membranes have been explored. Among these, the resorbable amniotic membrane has gained prominence due to its multiple beneficial properties. It helps maintain the structural and anatomical integrity of the regenerated tissues, thereby promoting effective healing [100].

The amniotic membrane is rich in stem cells, which can enhance tissue regeneration and repair processes. It also contributes to improved gingival wound healing, making it a valuable adjunct in periodontal procedures. Overall, the use of amniotic membranes in periodontal plastic surgery offers a promising alternative to traditional grafting techniques, providing better clinical outcomes with potentially reduced morbidity and enhanced regenerative capacity [1], as summarized in the following four tables, which indicate the specific use of the hAM, respectively, in periodontal surgery (Table 1), reconstructive surgery after tumor resection (Table 2), prosthodontic surgery (Table 3), and MRONJ (Table 4).

More recent experiences indicate good results in terms of root coverage, increased tissue thickness, and increased attached gingival tissue following the use of processed dehydrated allograft amnion, obtaining excellent aesthetic results in terms of texture and color match without postoperative discomfort and adverse reactions [117]. Grade II furcation defects have been successfully treated by the use of a demineralized freeze-dried bone allograft (DFDBA) with an amniotic membrane [118,119].

A study indicates that the hyperdry amniotic membrane is a promising intraoral wound dressing, demonstrating biological compatibility with oral tissues. Its potential as a clinical alternative for repairing the oral mucosa suggests that it may facilitate healing and tissue regeneration effectively. This could offer a valuable option in oral surgery and mucosal repair, potentially improving patient outcomes with a biologically acceptable and possibly more efficient material [120].

The use of a cryopreserved hAM as an interposed graft has been demonstrated, obtaining promising results in an experimental model of tension-free closure of oronasal fistulas in minipigs. This approach could potentially translate into improved surgical outcomes and simpler procedures for similar cases in humans [121].

The use of the hAM for closure of post-surgery defects in patient affected by oral submucous fibrosis has indicated that it is a biologically ideal graft for oral wounds and could be used for repair surgery for oral defects. It was found to be easy to use with good hemostatic properties and no complications [122].

Contemporary dental implant protocols recommend maintaining at least 1 mm of bone surrounding all aspects of the implant fixture. To achieve this, the concept of site preservation is frequently employed. Recently, resorbable amnion–chorion membranes have been introduced as a novel barrier for site preservation. Unlike traditional barriers such as cadaveric allografts, xenografts, and alloplasts, placental allografts are composed of immune-privileged tissue, conferring several advantages. They possess antibacterial and antimicrobial properties, reduce inflammation at the wound site, and provide a protein-enriched matrix that facilitates cell migration, thereby promoting optimal healing and regeneration [123].

The comparison between the application of hyaluronic acid and the hAM in clinical cases of gingival recession coverage and in intra-socket application for wound healing and bone regeneration has evidenced the potential usefulness of the hAM as well as hyaluronic acid in improving the postoperative sequelae following dental surgery in terms of pain, wound healing, and overall bone regeneration [124,125].

The most recent meta-analysis studies on gingival recession [126] and on MRONJ [23] have heightened the status of hAM as representing a feasible option and to show various beneficial properties in satisfactorily resolving the defects analyzed.

## 7. Conclusions

The amniotic membrane stands as a captivating and versatile biological material with a rich historical legacy in medicine. From its traditional applications in wound healing to its modern roles in regenerative medicine, the amniotic membrane represents a dynamic and evolving field of exploration [62]. As scientific knowledge advances, the therapeutic potential of this unique tissue is poised to play an increasingly pivotal role in shaping the landscape of medical interventions and patient care.

The amniotic membrane serves as an excellent scaffold for cellular proliferation and differentiation owing to its content of fibronectin, elastin, and various types of collagens [1,62]. One of its key advantages in allotransplantation or xenotransplantation is its lack of immunogenicity, reducing the risk of immune rejection. Additionally, the membrane exhibits multiple therapeutic properties, including promoting epithelialization, and possessing anti-inflammatory, antifibrotic, antibacterial, and antiangiogenic effects, attributes supported by the presence of specific bioactive factors.

The amniotic membrane notably facilitates epithelial cell migration, adhesion, and differentiation, making it an ideal substrate for supporting the growth and extending the lifespan of epithelial progenitor cells. Furthermore, due to these properties, the amnion has been widely utilized as a scaffold in tissue engineering research, contributing to advances in regenerative medicine and wound-healing applications [33,51].

The present review has offered a large body of evidence that confirms the use of the hAM as a valid tool to improve oral mucosa healing following therapeutic intervention, and in this context, hAM shows several properties and characteristics that proved to be useful in different fields of oral surgery.

The exceptional biological and biophysical properties of the human amniotic membrane, coupled with its wide availability and relatively low preparation, storage, and utilization costs, contribute to its superior performance compared to other grafts.

Further technological effort in hAM preparation and preservation could make this extraordinary instrument even more usable not only in the general field of wound healing, but also in the field of oral surgery.

### Unresolved Questions and Future Research Agenda

Despite promising clinical results, several important knowledge gaps remain regarding the use of the human amniotic membrane (hAM) in oral regenerative surgery. Addressing these unresolved questions will require well-designed, multi-center studies that combine clinical, biological, and patient-centered outcomes. Several questions outline priority areas for future investigation.

In particular, autogenous connective tissue grafting remains the gold standard for root coverage procedures, particularly in terms of long-term stability and aesthetic integration. A key question is whether modern hAM preparations can achieve non-inferiority compared to connective tissue graft (CTG), not only in short-term root coverage but also in maintaining results at 1–5 years. Beyond root coverage percentages, aesthetic outcomes such as gingival color match, tissue thickness, and patient satisfaction need systematic evaluation using validated scoring systems.

While the hAM has demonstrated benefits in soft tissue healing, its role in hard tissue regeneration remains less well defined. Critical questions include whether the hAM can enhance new bone formation in alveolar ridge preservation or guided tissue regeneration (particularly when membrane exposure occurs). Studies should examine whether the hAM influences implant-related outcomes such as stability, osseointegration, and marginal bone preservation at 1–3 years. Following this consideration, the potential for synergistic effects when combining the hAM with other biologics should be explored. Preliminary evidence suggests that adjunctive use with platelet concentrates (PRF, PRGF), enamel matrix derivatives, or even cell-seeded constructs may enhance regenerative outcomes. Systematic evaluation of such combinations could open new avenues for tissue engineering strategies that maximize both soft and hard tissue healing.

As has been evidenced in the present review, commercially available hAM products vary considerably in terms of processing (e.g., cryopreservation, lyophilization, dehydration, or irradiation) and handling characteristics. These differences likely influence bioactivity, sterility, and clinical performance. Comparative research should clarify which preservation methods retain growth factor activity and extracellular matrix integrity while maintaining ease of intraoral application. Furthermore, there is a need to define dosing strategies—such as membrane thickness, stacking of multiple layers, and expected resorption kinetics—under the unique conditions of the oral environment. Particular attention should be paid to the effects of sterilization methods, such as gamma irradiation, on both safety and clinical efficacy, since higher irradiation doses may reduce bioactivity and compromise clinical outcomes.

One of the strongest arguments for adopting the hAM in clinical practice will be its economic value relative to established alternatives. Formal cost–utility analyses, including quality-adjusted life years and cost per successful case, should be conducted in the context of root coverage procedures. These analyses would inform reimbursement models and guide decision-making for clinicians, patients, and healthcare systems.

## Figures and Tables

**Figure 1 ijms-26-08470-f001:**
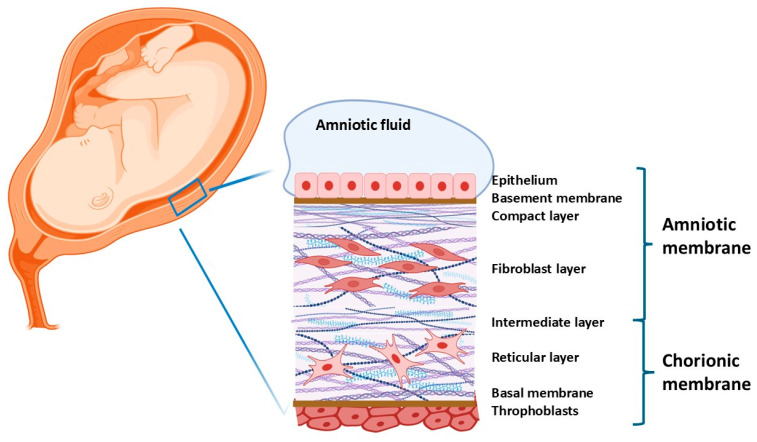
Anatomical structure of human amniotic and chorionic membranes in the placenta.

**Figure 2 ijms-26-08470-f002:**
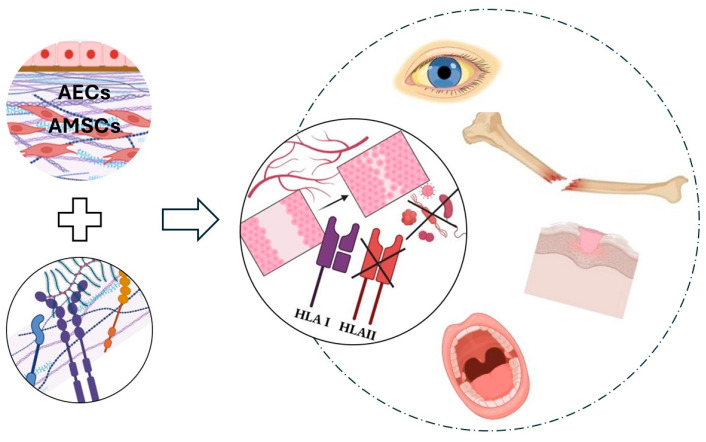
Properties of human amniotic membrane. The presence of stem cells and structural proteins, along with the secretion of growth factors and cytokines, enables the activation of multiple phenomena and beneficial effects, making the human amnionic membrane (hAM) an ideal biomaterial for medical applications.

**Figure 3 ijms-26-08470-f003:**
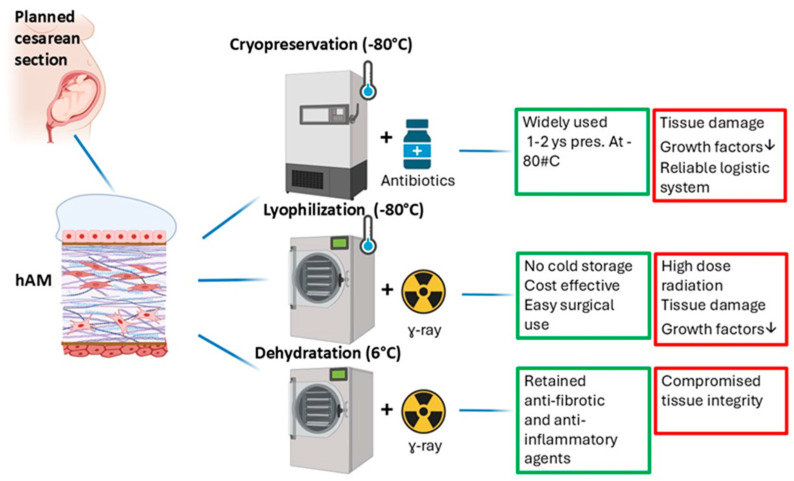
Schematic representation of hAM preparation methods and their respective advantages (green squares) and disadvantages (red squares).

**Table 1 ijms-26-08470-t001:** Use of amniotic membrane in periodontal surgery.

Year	Author	Patients	Indications	Treatment	Assessment Methods	Results
2018	Rehan et al. [101]	10	Gingival recession	Coronally advanced flap + platelet-rich fibrin (PRF) Coronally advanced flap + hAM	Plaque index; Gingival index; Bleeding on probing; Clinical attachment level; Depth of recession; Width of recession; Width of attached gingiva.	The hAM showed the better percentage of root coverage as compared to PRF.
2018	Kaur and Bathla [102]	15	Periodontal furcation defect	PRF + hAM hAM alone	Measurement of dental plaque index; Measurement of gingival index; Measurement of gingival recession depth; Measurement of pocket depth; Measurement of clinical attachment level.	All clinical and radiographic parameters showed statistically significant improvement at the sites treated with PRF and the amnion membrane compared to those with PRF alone.
2019	Temraz et al. [103]	22	Periodontal pockets	Open flap debridement + hAM Open flap debridement + demineralized bone matrix	Measurement of dental plaque index; Measurement of gingival index; Measurement of pocket depth; Measurement of clinical attachment level; Radiographic measurement.	Both the hAM barrier and demineralized bone matrix putty allograft provided significant improvement in clinical and radiographic outcomes after 6 months, yet no significant differences were noticed between them.
2020	Kadkh-oda et al. [104]	27	Healing of palatal donor site after free gingival graft surgery	hAM Only suture	Clinical assessment; Pain score.	Mean color match scores were higher in the hAM group than in the control group.
2020	Kumar et al. [105]	51	Gingival recession	Coronally advanced flap + hAM Coronally advanced flap alone	Measurement of clinical attachment level; Measurement of pocket depth; Measurement of recession width; Measurement of keratinized tissue width; Measurement of thickness of keratinized gingiva (TKG).	Intergroup comparison showed a non-significant difference in all settings except the TKG. The hAM was proven to help improve the TKG.
2021	Venkat-esan et al. [106]	50	Periodontal pockets	hAM + Biphasic calcium phosphate Collagen membrane + Biphasic calcium phosphate	Measurement of clinical attachment level; Measurement of pocket depth.	The hAM can be used as a barrier membrane, in conjunction with Biphasic calcium phosphate, and provides comparable results to a collagen membrane with Biphasic calcium phosphate.
2022	Agraw-al et al. [107]	20	Periodontal pockets	Open flap debridement + demineralized freeze-dried bone allograft + hAM Open flap debridement + demineralized freeze-dried bone allograft + collagen membrane	Measurement of dental plaque index; Measurement of gingival index; Measurement of pocket depth; Measurement of clinical attachment level; Radiographic measurement.	For all the clinical and radiographic parameters, no statistically significant difference was noted between both the groups.
2022	Nath et al. [108]	18	Gingival recession	Coronally advanced flap + hAM Coronally advanced flap alone	Measurement of width of attached gingiva; Measurement of clinical attachment level; Measurement of pocket depth; Measurement of width of keratinized gingiva; Measurement of length of gingival recession; Measurement of width of gingival recession.	Combined, a coronally advanced flap and the hAM have additional advantage in the outcome of periodontal therapy in the management of gingival recession.

**Table 2 ijms-26-08470-t002:** Use of amniotic membrane in oral reconstructive surgery after tumor resection or oral lesion excision.

Year	Author	Patients	Indications	Treatment	Assessment Methods	Results
2019	Akhlagi et al. [109]	9	Maxillomandibular bone defects following tumor surgery	Iliac crest bone graft + hAM Iliac crest bone graft + hAM + buccal fat pad-derived stem cells	Computed tomography image assessment	The mean increase in bone width was found to be significantly greater in the hAM + buccal fat pad-derived stem cell group
2022	Hazari-ka et al. [110]	15	Mucosal defect after excision of precancerous lesions	hAM	Clinical assessment of operability; Hemostatic status; Pain; Feeding situation; Epithelialization; Change in mouth opening; Mucosal suppleness and safety.	The hAM is a cost-effective material for immediate coverage of intraoral surgical defects.

**Table 3 ijms-26-08470-t003:** Use of amniotic membrane in prosthodontic surgery.

Year	Author	Patients	Indications	Treatment	Assessment Methods	Results
2021	Faraj et al. [111]	21	Alveolar ridge preservation	hAM Collagen membrane	Clinical assessment of ridge dimensions	Human amnion–chorion membrane or type 1 bovine collagen as the open barrier did not change healing.
2021	Babaki et al. [112]	28	Mandibular vestibuloplasty	hAM Acellular dermal matrix	Clinical assessment of relapses and healing	An acellular dermal matrix accelerates wound healing compared to the hAM.
2021	Gajul et al. [113]	10	Alveoloplasty	hAM Control	Assessment by Landry, Turnbull, and Howley Index	The hAM group showed an improved healing index for tissue color, bleeding on palpation, granulation tissue, suppuration, and overall healing.

**Table 4 ijms-26-08470-t004:** Use of amniotic membrane in medication-related osteonecrosis of the jaw (MRONJ).

Year	Author	Patients	Indications	Treatment	Assessment Methods	Results
2022	Canakci et al. [114]	5	MRONJ (zoledronic acid)	hAM	Clinical assessment	There was complete mucosal coverage in 4 patients
2022	Ragazzo et al. [115]	49	MRONJ (zoledronic acid, clodronic acid, ibandronic acid)	hAM	Clinical assessment	hAM seems to stimulate soft tissue healing and reducing pain perception in the postoperative period.
2022	Odet et al. [116]	8	MRONJ (bisphosphates, denosumab)	hAM	Clinical assessment	Of the lesions, 80% had complete or partial wound healing

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
