# Peer review of "The Human Amniotic Membrane: A Rediscovered Tool to Improve Wound Healing in Oral Surgery"

_ijms, 2025, doi:10.3390/ijms26178470_

Round 1
Reviewer 1 Report
Comments and Suggestions for Authors
Dear Editor,
Thank you for the opportunity to review this manuscript. The human amniotic membrane (hAM), a biological product with great potential to enhance oral surgery wound healing, was the focus of the current review. The review highlighted the membrane's complexity in terms of stem cell composition, extracellular matrix molecules, growth factors, and anti-inflammatory factors, all of which contribute to tissue regeneration, immune modulation, neovascularization, and infection management. Preservation methods of hAM and its use in periodontal, bone defect, and implant site regeneration are also covered.
Strengths:
- Comprehensive Coverage: The article goes into great detail about the biological properties of the human amniotic membrane, including its cellular makeup, molecular components, and natural biological functions for healing wounds.
- Novelty and Significance: Focus on oral surgery and the renewed interest in the application of hAM's utility in this field is a relevant and clinically useful topic. The addition of systemic factors like diabetes, smoking, and aging influencing wound healing broadens the clinical context.
- Data Presentation: The review appropriately delimits the immunomodulatory and regenerative roles of hAM, justifying the discussion with cytokines and growth factors such as TNF-α, IL-6, and IL-1β in inflammation and repair.
- Methods for Preservation: The application of treatment with cryopreservation, dehydration, and freeze-drying methods perfectly aligns with the realities of hAM preservation for clinical purposes.
While the study is well-designed and addresses an important area of regenerative medicine, several specific concerns need to be addressed to strengthen the manuscript for publication in the International Journal of Molecular Sciences.
- Abstract:
- A brief part should be added to the abstract to show significant conclusions and clinical relevance better.
- Structure and Flow:
The article would benefit from clearer subheadings and improved logical sequence. For example, sectioning the biological properties, preservation methods, and clinical applications clearly would enhance the readability of the paper.
- Depth of Clinical Application:
While the biological background is well covered, more focus on clinical evidence, like outcomes from human trials or series of cases in oral surgery, would make the review more translatable.
- Integration of Recent Literature:
Add to the review the recent research up to 2024-2025, mainly from high-impact journals, to provide contemporary evidence regarding hAM in tissue engineering and oral surgery.
- Clarification on Immunogenicity:
The mention of HLA class II molecules and immune modulation in pregnancy is relevant. It should, however, describe how specifically this translates to clinical safety of oral administration in terms of rejection and immune tolerance.
- Figures:
The addition of schematic figures demonstrating hAM's biological activities and a table of clinical trials or outcomes in oral surgery would be a very positive addition to understanding.
Author Response
REFEREE 1
The authors thank the reviewer for appreciation of the article.
1. The abstract has been slightly modified to address the review suggestion.
2. It is our opinion that a greater fragmentation into subchapters would not be useful. In this regard, we allow ourselves to present the list of sections and subsections of the work, which we believe to be quite comprehensive.
- Human Amniotic Membrane Structural Complexity
- The hAM in tissue repairing
3.1. Stem cell characteristics of amnion‑derived cells
4 hAM multifaceted properties
4.1. Biomechanical Properties
4.2. Epithelialization
4.3. Inhibition of Inflammation
4.4. Angiogenesis
4.5. Inhibition of scarring
4.6. Lack of Immunogenicity.
4.7 Antimicrobial / Antiviral properties
4.8. Aesthetic properties4.9. Reduces pain at the site of application
- Preparation and utilization of hAM
5.1. Cryopreservation
5.2. Lyophilisation
5.3. Dehydration
6.The hAM in oral surgery
3. More clinical evidence have been added at the end of the section 6 “The hAM in oral surgery” in form of Tables
4. No recent paper up to 2024-2025 about hAM has been found further those cited in the work. If the referee would like to suggest specific papers, we will be happy to add them to the article.
5. Further sentence to extend the topic of HLA class II has been added in the text (see pag. 7, lines: 317-322 and 327-334).
6. Four different tables have been added to focus some clinical trials and outcomes in oral surgery
Reviewer 2 Report
Comments and Suggestions for Authors
The manuscript titled "The Human Amniotic Membrane: a rediscovered tool to improve wound healing in oral surgery" provides a comprehensive review of the biological, biomechanical, and therapeutic properties of human amniotic membrane (hAM) with a focus on its applications in oral surgery. The review covers stem cell properties of amnion-derived cells, multiple biological activities (including epithelialization, angiogenesis, anti-inflammatory effects, antimicrobial properties, and scar reduction), various preservation and processing techniques (cryopreservation, lyophilization, dehydration), and clinical experience in periodontal and oral reconstructive procedures. The authors support their discussion with references to basic research, preclinical studies, clinical case reports, and meta-analyses.
However, this literature review is too long, with redundancy and inconsistent terminology. With refinement, it could be a valuable reference.
- The manuscript's long sentences make it hard for readers to follow. Properties like anti-inflammatory and antimicrobial are repeated in multiple sections without clear integration. Clinical sections blend basic science explanations and surgical technique descriptions; separating them could improve clarity.
- The authors should standardize all terminology and abbreviations throughout.
- While the descriptions of cryopreservation, lyophilization, and dehydration are detailed, they are presented in dense text. Comparing the advantages, disadvantages, and clinical impact of each method is difficult. Recommendation: A comparative table is needed that summarizes the storage requirements, preservation of biological activity, risks, and logistical considerations of each method.
- The strength of clinical evidence is not always specified; in some cases, it is unclear whether claims are based on in vitro studies, small case series, or randomized clinical trials. Therefore, for key claims, specify the level of evidence (e.g., “supported by randomized clinical trials,” “based on preclinical animal studies”).
- For the Conclusion section, research gaps are not explicitly discussed. The authors should reframe conclusions to emphasize both current evidence and unresolved questions.
Author Response
The authors thank the reviewer for appreciation of the article. We believe the complexity of the review topic is the main reason for its length. However, we have tried to minimize redundancy.
1.We've tried to improve the writing style to make the article more understandable. Changed or adjusted sentences have been highlighted in yellow in the revised copy.
- An effort has been made to standardize all terminology and abbreviations. We hope we have succeeded.
- A schematic figure about technical methods for hAM preservation has been added.
- Four tables summarizing clinical studies has been added.
- The conclusion section has been extended with a further paragraph, where the unresolved questions or further research needed are highlighted
Round 2
Reviewer 2 Report
Comments and Suggestions for Authors
the manuscript is improved